# Archaeological expansions in tropical South America during the late Holocene: Assessing the role of demic diffusion

**Jonas Gregorio de Souza**[1]*, **Jonas Alcaina Mateos**[1], **Marco Madella**[1,2,3]

**1** Department of Humanities, Culture and Socio-Ecological Dynamics group (CaSEs), Universitat Pompeu Fabra, Barcelona, Spain, **2** Institució Catalana de Recerca i Estudis Avançats (ICREA), Barcelona, Spain, **3** School of Geography, Archaeology and Environmental Studies, The University of the Witwatersrand, Johannesburg, South Africa

* jonas.gregorio@gmail.com

**Data Availability Statement:** All relevant data are within the manuscript and its Supporting Information files.

## Abstract

Human expansions motivated by the spread of farming are one of the most important processes that shaped cultural geographies during the Holocene. The best known example of this phenomenon is the Neolithic expansion in Europe, but parallels in other parts of the globe have recently come into focus. Here, we examine the expansion of four archaeological cultures of widespread distribution in lowland South America, most of which originated in or around the Amazon basin and spread during the late Holocene with the practice of tropical forest agriculture. We analyze spatial gradients in radiocarbon dates of each culture through space-time regressions, allowing us to establish the most likely geographical origin, time and speed of expansion. To further assess the feasibility of demic diffusion as the process behind the archaeological expansions in question, we employ agent-based simulations with demographic parameters derived from the ethnography of tropical forest farmers. We find that, while some expansions can be realistically modeled as demographic processes, others are not easily explainable in the same manner, which is possibly due to different processes driving their dispersal (e.g. cultural diffusion) or problematic/incomplete archaeological data.

## Introduction

Following the successful colonization of the globe by our species, new waves of human expansion happened during the Holocene, reshaping cultural, linguistic and genetic landscapes worldwide [1–3]. Such expansions may have been triggered by the emergence of food production economies and associated increases in population growth rate [4]. The demographic and technological advantages offered by the onset of agriculture as drivers of Holocene cultural expansions are also supported by the distribution of the largest language families of the world, as summarized in the Farming-Language Dispersal Hypothesis (FLDH) [5]. According to the FLDH, the uneven distribution of language families in the world is explained as a consequence of population growth of farmers and their subsequent expansion from early domestication

**Funding:** JGS has been funded by a MSCA individual fellowship (Grant 840163, EU Horizon 2020). The funders had no role in study design, data collection and analysis, decision to publish, or preparation of the manuscript.

**Competing interests:** The authors have declared that no competing interests exist.

centers to spread their language, culture and genes over vast areas. Later work has been concerned with analysing the timing, pace and routes of human expansions using computational and statistical approaches to archaeological data, exploring the regional distribution of $^{14}$C dates and simulating scenarios of demic and cultural diffusion based on demographic parameters [6–10].

The Americas have never figured as prominently as Eurasia, Africa or Oceania in such debates [11]. South America has been notably absent, with the few published examples lacking analyses of the gradients in archaeological $^{14}$C dates or development of testable models [12–14]. In part, this is due to the substantially smaller and less reliable dataset when compared to what is available for Eurasia. Another drawback is the commonly accepted idea that continent-wide expansions did not occur in tropical South America due to environmental constraints or peculiar socioeconomic trajectories [15–17]. However, the vast territories occupied by the largest language families of South America contradict that claim, suggesting that sizable demographic expansions may have taken place in pre-Columbian times [18,19]. Further support is offered by the distribution of archaeological cultures that emerged and expanded over the last 5000 years (Fig 1) [20–22].

Here we examine four archaeological phenomena (Fig 1): (1) the Saladoid-Barrancoid series and related cultures, including Pocó-Açutuba and other phases belonging to the Incised Rim tradition [21]; (2) the Arauquinoid culture and related phases included in the Incised-Punctate tradition [23]; (3) the Tupiguarani tradition [20]; and (4) the closely-related Una, Itararé and Aratu traditions [24,25]. In most of the regions they settled, those archaeological cultures introduced the cultivation of domesticated plants, marked the transition to more permanent settlements, and diffused practices of landscape modification including the formation of anthropogenic dark earths (ADE) [26,27]. In summary, they spread an economic model conventionally called "polyculture agroforestry", combining extensive cultivation of domesticated plants with the management of useful and semi-domesticated species in enriched forests, rather than large-scale clearance for monoculture [16,17,22,28,29]. Those practices are fundamentally different from the slash-and-burn agriculture that is nowadays ubiquitous in the Amazon, and from the cereal agriculture that spread with the Neolithic in Eurasia, but represent a model of sustained cultivation adapted to tropical forest environments [16]. For simplicity, in the remainder of this paper, the term "agriculture" in the Neotropics must be understood as implicitly referring to polyculture agroforestry.

Some of the archaeological expansions that we examine are traditionally assumed to have spread by means of demic processes [30]. Others have been suggested to reflect cultural diffusion or trade networks [12,31]. Whatever the case, correlations between the archaeological cultures in question and the largest language families of tropical South America have long been proposed [32,33]. For those reasons, archaeological expansions in lowland South America during the late Holocene may represent a phenomenon comparable to other agricultural waves summarized in the FLDH, at the same time that they provide a valuable contrast due to their multifocal nature and the distinct subsistence basis represented by polyculture agroforestry [17]. In fact, it has been suggested that the expansion capacity of early agricultural systems was greatly determined by the type of economic "package"—whether it included tree and root crops (as in lowland South America) or cereals and herd animals [34]. With that in mind, we aim at answering the following questions: (1) Can the archaeological expansions that took place over lowland South America in the late Holocene be modeled as processes of demic diffusion? (2) What sets of demographic parameters better predict the rhythm of expansion of each archaeological culture?

Following, we provide a brief contextualization for the reader unfamiliar with the archaeology of lowland South America. We then proceed to explore the spatial gradients in the

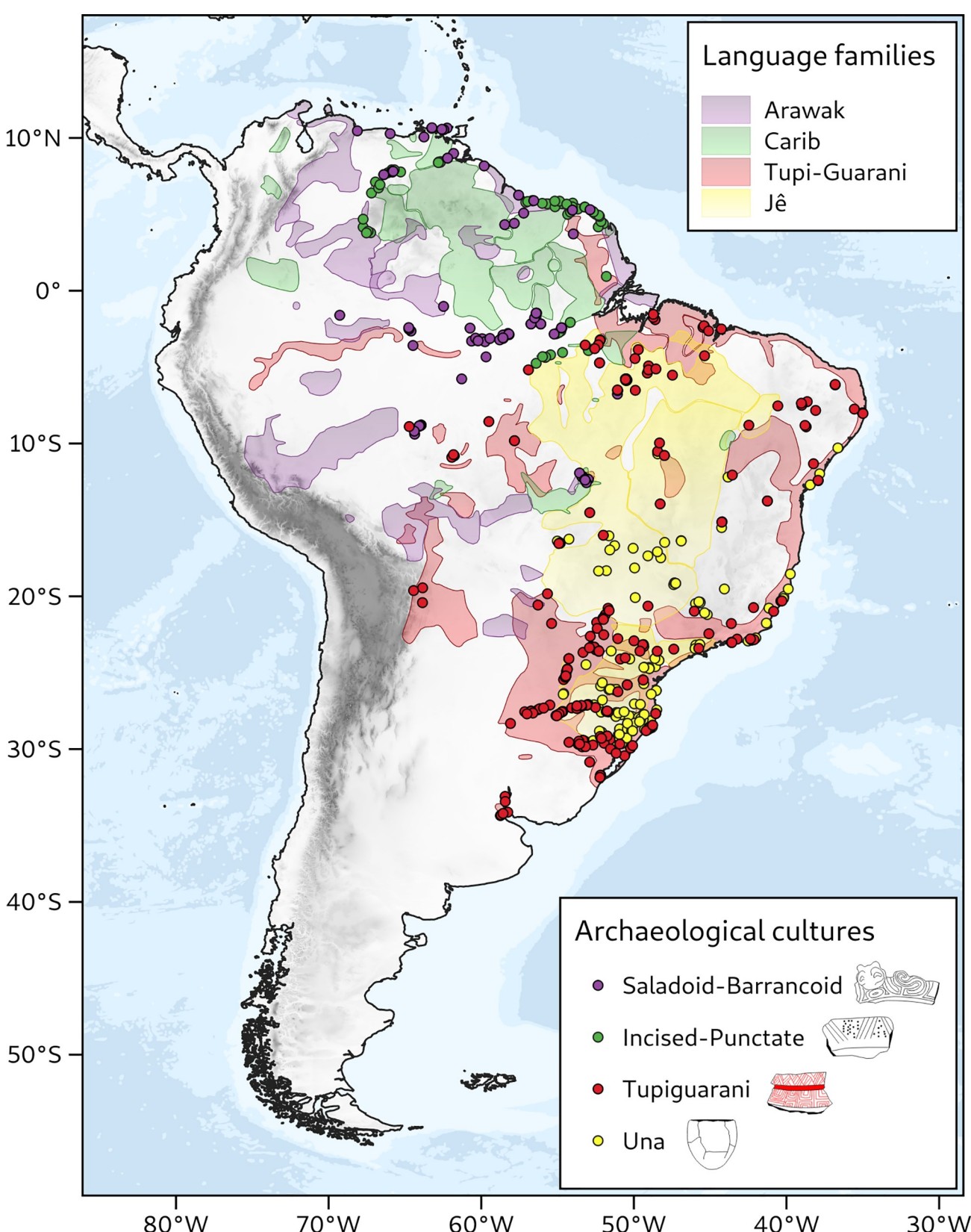

**Fig 1. Distribution of dated sites of four late Holocene archaeological cultures in tropical South America superimposed on the distribution of the largest language families of the continent.**

chronology of each archaeological culture by performing regression analysis of radiocarbon dates and distances from hypothetical centers of origin, using established methods for determining the most likely geographical origins and diffusion rates [6,8,35]. To further assess the feasibility of demic diffusion as the mechanism behind the archaeological phenomena under consideration, we perform agent-based simulations of human expansions based on realistic demographic parameters and dynamics of movement, adapting procedures commonly employed in the literature [9,36] to the reality of tropical forest agriculturists.

## Brief summary of the archaeological case studies

**Saladoid-Barrancoid.** Under the label *Saladoid-Barrancoid*, we group, beyond the Saladoid and Barrancoid series of the Orinoco, also the central Amazonian Pocó and Açutuba complexes, which share ceramic stylistic attributes, are of similar age, and must be considered variants of the same phenomenon [21,37,38]. In addition to similarities in material culture, the Saladoid-Barrancoid unity is apparent in settlement patterns, with early presence of ADE and circular village plans throughout the Amazon [21,39,40]. Ceramic phases previously included in the Incised Rim or Incised-Modelled tradition represent later variants of this horizon [33,39,41,42].

The spread of Saladoid-Barrancoid material culture can be traced to the Orinoco ca. 4600 cal BP, expanding over most of the Amazon by ca. 3400–2900 cal BP, except for the southern limits of the basin, which were settled ca. 1000 cal BP [39,43]. Saladoid ceramics can also be found outside South America, having spread as far as Puerto Rico and Hispaniola–a maritime expansion that is beyond the scope of this article [44].

Saladoid-Barrancoid occupations are associated with initial ADE formation, marking the first settlement of areas where hunter-gatherer sites are scarce or absent [21]. Their dispersal may have been triggered by the successful development of agriculture following an interval of drought in the mid-Holocene [16,45]. That hypothesis is supported by archaeobotanical evidence, with maize (*Zea mays*), manioc (*Manihot esculenta*), yam (*Dioscorea sp.*) and other cultigens recovered from sites in the Orinoco [46]. Maize is also present in central Amazon, in association with disturbance taxa and edible species such as *Bactris* sp. palms [47].

**Incised-Punctate.** The Incised-Punctate horizon encompassesseveral late eastern Amazonian cultures whose ceramic decoration has prototypes in the earlier Arauquinoid style of the Orinoco [38,48–51]. The distribution of Incised-Punctate sites overlaps in part with the Saladoid-Barrancoid horizon, but Incised-Punctate sites are later and geographically more restricted (Fig 1). They originated in the Orinoco ca. 1600 cal BP, spreading to the Guianas coast by ca. 1300 cal BP and reaching the lower Amazon at ca. 800 cal BP [29,38].

Incised-Punctate settlement patterns and site architecture are diverse. Along the Guianas coast, settlements included habitation mounds surrounded by complexes of agricultural raised fields [51,52]. In eastern Amazon, some of the largest ADE sites are found, and their density is also one of the highest [53,54]. The agricultural base of the Incised-Punctate tradition is clear, with maize present from the Orinoco to French Guiana, where it occurs in raised fields alongside squash (*Cucurbita* sp.) and manioc [43,52]. In eastern Amazon, the spread of Incised-Punctate settlements coincides with ADE formation and cultivation of maize, manioc and squash [29].

**Tupiguarani.** Apart from the Saladoid island expansion to the Antilles, the Tupiguarani tradition was the only archaeological culture under consideration that spread outside of the Amazon. Sites are found in the central Brazilian savanna (*cerrado*), semiarid zones of northeastern Brazil, the Atlantic coast, and subtropical forests–reaching as far south as the pampas of Buenos Aires [20,55]. Contrasting with their territorial extent is the remarkable

homogeneity in material culture, with the same repertoire of ceramic shapes and decoration everywhere in the Tupiguarani territory [56]. Settlement patterns also exhibit broad similarities, with ADE sites corresponding to single communal houses or villages located along navigable rivers [13,20,57].

The chronology of the Tupiguarani expansion is problematic (see Methods). Earlier attempts to situate its origin in central or southwestern Amazon have been reconsidered and today eastern Amazon is suggested as the most likely cradle of the Tupiguarani dispersal [58]. Excluding unreliable evidence, the Tupiguarani tradition appears to have spread from the basins of the Xingu and Tocantins rivers from ca. 2400 cal BP, rapidly reaching the Paraná basin at ca. 2200 cal BP and the Atlantic coast at least ca. 1600 cal BP [13,58,59].

Tupiguarani subsistence relied on cultivated plants and forest management. Microbotanical remains of maize and beans (*Phaseolus* sp.) have been identified in the southernmost sites, where maize also appears in the pollen record close to the time of Tupiguarani arrival [60,61]. Consumption of maize has been attested by stable isotope data from human skeletal remains [62]. In addition, secondary vegetation in the surrounding of Tupiguarani settlements has been inferred from anthracological analyses [63].

**Una and related phases.** All the archaeological cultures discussed so far originated in Amazonia. In the central Brazilian savanna (*cerrado*), a separate focus of agriculture and ceramic diffusion is represented by the Una tradition [24]. In this paper, we group under the label *Una* also other related, non-Amazonian traditions that are spatially and temporally contiguous and have their roots in central Brazil: the Taquara-Itararé tradition of the subtropical highlands of southern Brazil and the Aratu tradition of the *cerrado* and coast [24,25,55,64,65].

The origins of Una expansion can be traced to ca. 3700 cal BP in the rock shelters of the Paraná and São Francisco headwaters, whence it spread throughout the *cerrado* by ca. 2400–1800 cal BP, settling most of the southern highlands and Atlantic coast ca. 1000 cal BP [24,65–67]. The earliest period is marked by a common settlement pattern, with the occupation of rock shelters in the *cerrado* and southern highlands [24]. Such a pattern is quite distinct from the tropical forest traditions reviewed so far. When compared to the Amazonian cultures, Una material culture is also characterized by its simplicity. In later periods, however, larger and more permanent settlements appeared, including pit house villages associated with other types of earthen architecture like mounds and enclosures in the southern highlands [68,69] and ring villages in the *cerrado* and Atlantic coast *[64]*.

In spite of the distinctiveness in its settlement patterns, the Una expansion has also been sustained by an agricultural base, as evidenced by the presence of maize in storage facilities since the earliest occupations in rock shelters, together with other domesticated plant remains like manioc and beans in later periods [70,71]. Microbotanical evidence of maize, beans and squash has also been recovered from southern Brazilian pit houses, where maize cultivation is further attested in the pollen record [72–74]. Importantly, ancient genomes point to a dispersal of certain lineages of maize in association with central Brazilian ring villages [75].

**Possible correlations with language families.** Direct connections between material culture, language and ethnicity in Amazonia are not without problems. Nevertheless, persistent associations of ceramic styles, settlement patterns and site architecture have been shown to pervade the Amazonian past, with strong links to historically recorded populations [33]. For example, the spread of the Saladoid-Barrancoid culture has been hypothesized for some time to reflect the expansion of Arawak languages [32,55,76,77]. Similarly, the Incised-Punctate expansion has been suggested to coincide with the spread of languages of the Carib family [32,78]. As for the Tupiguarani tradition, the very name given to the archaeological culture reflects the fact that its sites almost perfectly mirror the historical distribution of the Tupi-Guarani language family, one of the most widespread of South America [56,79]. Finally, Una-

related traditions have long been associated with the Macro-Jê language stock, of ancient dispersal in central Brazil [55,80,81]. Although we do not fully endorse perfect correspondences between material culture and language, we admit that, if demic expansions could be demonstrated to lie behind the archaeological phenomena in question, their association with language expansions in South America would be given further weight, fitting the expectations of the FLDH.

## Materials and methods

### Compilation of radiocarbon dates

Radiocarbon dates and associated coordinates were compiled from the published literature, academic theses and reports (S1 Table). Information as complete and reliable as possible was recorded pertaining to cultural affiliation, site type, and potential problems with the date, when available. Dates with standard errors larger than 200 years were excluded. Although more precision would be desirable, retaining dates with even lower standard errors would lead to the exclusion of many dates published in the 1960s and 1970s. Using lower thresholds could also lead to the arbitrary removal of dates that are crucial for determining first arrival times in many regions [82]. Dates were calibrated using the ShCal13 curve [83]. Given that some sites are located north of the equator, an alternative approach would be to use the northern hemisphere curve for those cases, or to use a mixed curve to account for the mixing of atmospheric carbon from the two hemispheres caused by the annual displacement of the ITCZ over South America [84]. Considering that the difference between the southern and northern hemisphere curves is in the order of ca. 15 years, we used, for simplicity, ShCal13 for all cases, as it has been done in previous works that used radiocarbon dates from different parts of South America [45,85].

The dataset was further filtered for unreliable dates whose association with the archaeological context or cultural affiliation was questionable. Dates that were considered unreliable by the researchers who obtained them or that were found to be unreliable by later reviews were also excluded. Nevertheless, not all cases could be resolved in such a manner and therefore, given the scope of the dataset and the lack of consensus on cultural taxonomies in the archaeology of lowland South America, S1 Table also lists the sites excluded from the filtered dataset. We annotated the reasons for removal so as to make our decisions as transparent as possible. In what follows, we explain our choices regarding the most important controversies.

**Saladoid-Barrancoid.** The main debate for this archaeological tradition rests on the distinction between a "long chronology" and a "short chronology" of the Saladoid series in the Orinoco. The long chronology places the origins of Saladoid ceramics at ca. 4500 cal BP, whereas the short chronology delays its origins at ca. 2800 cal BP. The Barrancoid style, on the other hand, is commonly accepted to have begun ca. 3000 cal BP. Given the lack of consensus, and supported by the recently obtained dates for the Saladero phase, which show that the development of that ceramic style fits poorly in the short chronology [46], here we provide an analysis using the long chronology. Indeed, it is easier to conciliate the long chronology if the early dates for Pocó-Açutuba sites in central and lower Amazon [21] are considered. Nevertheless, for the sake of methodological integrity, we decided to run separate regression analyses and simulations also adopting the Orinoco short chronology. For running these models, we removed the earliest dates reported by Roosevelt [38] for La Gruta ceramics.

**Incised-Punctate.** Sites of the early Aristé phase in the Guianas, originally classified as Incised-Punctate, have not been included in our analysis, since Aristé ceramics are no longer considered part of this tradition [86,87] Their dates are also peculiarly early when compared with the chronology of the Orinoco and lower Amazon. Koriabo sites, however, have been

included in spite of doubts about their broader affiliation [88], since the ceramics show much clearer affinities with lower Amazonian complexes [89,90]. Regardless of our decision, the Koriabo culture is recent, and excluding its dates from the analysis would not substantially affect the results.

**Tupiguarani.** The Tupiguarani chronology is surrounded by controversies. The putative epicenter of the Tupi language stock in southwestern Amazonia, near the present-day state of Rondônia, has led to a search for early dates in that region [91–93]. The dates published by Miller [93], some earlier than 5000 cal BP, are often cited as confirmation of a southwestern Amazonian origin [14,56]. Beyond the contextual problems that such dates pose, the cradle of the Tupi-Guarani language family (not the Tupi stock) is clearly located in eastern (not southwestern) Amazon [19,94,95]. Another difficulty in Tupiguarani chronology is represented by recently published dates from the coast of Rio de Janeiro state, inexplicably early when compared to other Tupiguarani occupations outside of Amazonia [59,96]. In light of those numerous problems, we used a conservative approach and excluded all the dates for which some objection could be raised, thus removing outliers from southwestern Amazon [91,92,97], central Brazilian rock shelters [98], and the coast [59].

## Regression analysis

To assess the relationship between chronology and geographical dispersion, we followed the well-established practice of performing linear regression of dates versus great-circle distances from hypothetical origins [6,8,35,82,99,100]. Ordinary least squares (OLS) is commonly adopted for such ends, with regression of dates versus distances following the assumption that the former variable contains most of the error, whereas the latter is, in principle, known with certainty [8,82,101]. Given that *de facto* travelled distances were probably not linear, this assumption is problematic. Here, we prefer the alternative of using reduced major axis (RMA) regression. While OLS assumes that the independent variable is measured without error, RMA assumes a symmetrical distribution of error between both variables [99,102]. RMA has been shown to perform better at identifying the true relationship between two error-prone variables, being robust to outliers and preserving the underlying slope even when the dataset is reduced [103].

We adopted the standard practice of measuring great-circle distances from the site with the earliest date discovered so far, which is assumed to be the probable center of origin [6]. This procedure, however, may be inadequate in certain cases, such as when the earliest date does not correspond to the beginning of the expansion [36]. In addition, given potential research biases in the location of the earliest dates, other locations should be considered [35]. Thus, although we primarily rely on the regression of dates and distances from the earliest site, we also tested for other possible origins by iterating over all sites.

The inclusion of all available dates per site for line-fitting has been considered problematic, given the distortion caused when early sites also have more recent dates [101]. The most commonly employed solution is to include only the earliest date available for each site, assuming they reflect the event of first arrival in a given location [6,100,101]. Additionally, to account for biases in the regional distribution of dates, some form of spatial filtering is frequently employed. One approach is to cluster sites into spatial bins defined by regular distance intervals measured from the center of origin [7,35,99]. This approach is more adequate for estimation of front speeds due to the fact that, in theory, the number of sites is correlated with time and distance from origin for a population expanding at a constant rate [99]. The bin size must offer a compromise between maximizing the number of sites included in the analysis and ensuring a minimum distance for the identification of underlying trends [35]. Here, we tested

a range of spatial bin widths from 100 to 500 km in intervals of 100 km, excluding those cases where fewer than 5 dates were retained after filtering.

One of the difficulties inherent in analyzing radiocarbon dates as a function of distance is the use of point values for dates. The modal values of uncalibrated $^{14}$C dates and the medians of calibrated probability distributions have commonly been utilized for that end [6,99,103]. When using calibrated dates, however, point estimates are inadequate when the probability distribution is irregular [6]. One solution is to assess the robustness of the regression by boot-strapping [6,99,102], which is the practice adopted here. For each geographical origin considered, we performed repeated regressions (n = 999), each time drawing single calendar year values from the radiocarbon dates with a probability given by the calibrated distribution. We report the averages of the correlation coefficient, intercept and slope, with a 95% confidence interval given by the bootstrapping procedure.

**Simulations.** We have designed an agent-based model to simulate demographic expansions based on parameters of growth and dispersal derived from the ethnography of Amazonian farmers [104–111]. The model was implemented in Python (S1 Appendix) and its architecture is similar to that of other simulations of human expansions [9,36]. We attempted, however, to reproduce the dynamics of village fission, short-distance moves and long-distance migration with greater realism considering the peculiarities of neotropical agriculturist populations. The model comprises individual agents, each corresponding to a village, who move on a bidimensional grid representing South America. For accurate calculation of distances, we used Albers Equal Area Conic projection and square grid cells with 10 x 10 km. Considering the average catchment area and inter-village distance reported for ethnographic Amazonian populations, the selected cell size was found to be the most adequate for representing individual village movement and fission.

**Model space.** Models of demographic waves of advance often assume homogeneous spaces where expansion occurs equally in any direction. In reality, human expansions unfold in a discrete manner and are strongly influenced by the environment, following more attractive areas and avoiding inhospitable regions or difficult terrain [7]. For that reason, we included a value for landscape attractiveness in the model grid. The measure of attractiveness was based on (i) elevation, (ii) slope, (iii) distance from rivers, (iv) net primary productivity, and (v) soil quality. We consider that the selected parameters are the most relevant for the practice of agriculture in the tropics, as well as being good predictors of archaeological site occurrence [27]. Elevation and slope were derived from the Shuttle Radar Topography Mission (SRTM) (https://eros.usgs.gov) and resampled to the grid resolution. Distance from rivers was calculated using HydroSheds data, selecting features with flow accumulation over 100,000 cells [112]. Net primary productivity, a powerful predictor of human population density [113,114], was calculated using the Miami Model [115] with late Holocene precipitation and temperature data derived from the Community Climate System Model (CCSM) [116,117]. For soil quality, we adopted the classification of nutrient availability from the harmonized word soil database [118]. With those parameters, we modeled ecological niches for each archaeological culture using maximum entropy algorithms implemented in MaxEnt [119–121]. Maximum entropy models are commonly used for predicting species distribution from presence-only data and have seen a number of archaeological applications [27,122–125]. For presence data, beyond the dated sites, we included coordinates of sites compiled from the Brazilian National Register of Archaeological Sites and relevant literature [126]. Separate models were built for each culture, and the standard output of MaxEnt, a cloglog-transformed probability of presence, was utilized as a layer of environmental suitability in each simulation, influencing the agents' decision to move to a cell (S1 Fig). To further restrain the movement of the agents, we used maximum training sensitivity plus specificity [127] as a threshold value for the ecological niche of

each culture, beyond which they were not allowed to settle. This procedure was found to reproduce with realism the observed distribution of each archaeological culture.

**Parameters and behavior of agents.** Models of demic expansion often reproduce the reaction-diffusion formula [35,82,128]:

$$\partial N/\partial t = aN(1 - N/K) + D\nabla^2 N \qquad (1)$$

Where $N$ is the population, $a$ is the intrinsic population growth rate, and $D$ is a diffusion coefficient or dispersal kernel for a given period of time ($t$). The formula incorporates the logistic function of population growth constrained by carrying capacity ($K$). Later corrections of this model recognize that human dispersal is discrete in time, since migration does not begin at an individual's birth, and usually include a time delay given by the length of a generation [7,8,101]. The implementation of demic diffusion models in computer simulations is minimally accomplished by a sequence of (1) population growth, (2) fission and (3) migration, by means of which a growing population is redistributed to its surrounding space [36,128,129]. Here, we adopt a similar concept.

Growth rate ($a$) is kept constant in all simulations. We have chosen the value 0.025 (2.5%) yr$^{-1}$, which is lower than the value calculated for expanding populations of agriculturists in Eurasia (0.029–0.035) [7,8] and is very close to the growth rate calculated from bioanthropological data for the Neolithic [9]. In fact, the growth rates of many Amazonian groups have been shown to stabilize at ca. 2.8% after recovery from post-contact population declines, with many of them exhibiting even higher growth rates [130]. We have chosen a lower value to reflect the fact that population expansions in our model did not occur in a vacuum and following the assumption that growth rates in polyculture agroforestry systems may have been lower than in Eurasian-type agropastoral systems.

The following parameters have been selected to model population growth and movement (Table 1):

*Maximum population density*: In logistic models, growth is controlled by carrying capacity ($K$), the limit beyond which resources are not sufficient to sustain the population. In spite of its theoretical usefulness, $K$ is challenging to implement. Here, we use the related concept of $K^*$, the observed maximum population density or equilibrium population [131], without making theoretical assumptions about limitations to growth. For agriculturists, maximum population density derived from anthropological and genetic data is 1.28 individuals km$^{-2}$ [9,36]. In Amazonia, modern observed densities exhibit much variation, but are generally lower. Some populations exhibit low densities of 0.1–0.2 individuals km$^{-2}$, while others are in the range of 0.8–1 individuals km$^{-2}$ [108,111,132,133]. Even at the lower bound, those densities are still higher than those observed among tropical hunter-gatherers [134]. In the past, the tendency was to project low densities of 0.2–0.3 individuals km$^{-2}$ for pre-Columbian Amazonia, although some environments, like floodplains, were considered to have sustained larger populations

**Table 1. Parameters for the simulations with respective initialization values in the genetic algorithm.**

| Parameter | Range | Unit |
|---|---|---|
| Growth rate ($a$) | 2.5 (constant) | % |
| Maximum population density ($K^*$) | 20–100 | individuals 100 km$^{-2}$ |
| Catchment radius | 10–30 | km |
| Leap distance | 0 or 150–250 | km |
| Fission threshold | 50–300 | individuals |
| Permanence | 10–30 | years |

[135,136]. Modern estimates suggest larger pre-Columbian populations, with 1–2.5 individuals km$^{-2}$ projected for broad regions of Amazonia, and even higher in some specific areas [125,137,138]. We initialized the models with maximum density values of 0.2–1 individuals km$^{-2}$.

*Fission threshold*: The phenomenon of village fission has been extensively discussed in archaeology and anthropology as a mechanism of stress alleviation in the absence of hierarchy [139]. Regardless of its causes, village fission is an important driver of demic expansion and is commonly implemented in simulations by moving a percentage of the population to the neighboring cells [8,9,129]. We established a limit for village population, beyond which fission would happen. Fission thresholds are variable, with a critical value of ca. 158 individuals deduced from a range of archaeological and anthropological records [140]. In Amazonia, however, prehistoric settlements with populations calculated in the lower thousands have been described [64,125,137,141]. Among modern Amazonian agriculturists, villages can be small, averaging 30–60 individuals in some groups [105,106], or reach 250–350 individuals before they fission in densely populated regions like the Upper Xingu [39,142]. Many populations are in between, living in villages of ca. 145 individuals, close to the average described in other parts of the world [111,139,140]. Village size may depend on a number of factors that are beyond the scope of our model, such as the benefits of large populations for village defense in regions where warfare was intense [132]. In the genetic algorithm (see Model Evaluation below), we initialized the models with fission thresholds of 50–300 individuals.

*Catchment radius*: This parameter reflects the territory exploited by a village and, as such, also controls the average inter-village distance. Demic diffusion models often proceed by dispersing, at each step, the population from each cell to the adjacent cells. However, we consider that a more realistic approach is to take catchment into account. New villages formed as a result of fission should move far enough to establish their own catchment area, and the same applies for cases of village relocation. In the Amazon, most agriculturists have their gardens within 5 km of the village, but a wider catchment is exploited for hunting and gathering [108,110,143]. Some Amazonian villages are located far apart, with distances of up to 50–80 km [105,108]. Villages have been reported to move 20–25 km when relocating, confirming the importance of maintaining a reasonable distance for the establishment of new exploitation territories [108,110,111,142]. In regions that are more densely settled, however, village territories span only 5–10 km [39]. This is probably due to the gradual overpopulation of the landscape leaving less land available to move. We initialized the models with catchment radii of 10–30 km.

*Permanence*: Population expansions in Amazonia are driven by a combination of village fission and village relocation. Most Amazonian agriculturists practice two modes of village relocation: (i) frequent, short-range moves of less than 1–2 km to its immediate adjacency, usually in order to be closer to newly opened gardens, and (ii) less frequent, long-range moves provoked by a variety of factors including warfare and house deterioration [108,110,111,132]. Given the resolution of the grid (10 x 10 km), we did not simulate the first type of movement, but included maximum permanence time as a parameter to control long-range moves. Some Amazonian groups are very mobile [106,111]. In general, however, populations depending on agriculture may remain for over 10 years in the same location [110,132,144]. In densely settled areas like the Upper Xingu, village relocation (not considering fission) is even rarer [39]. Archaeological settlement patterns in Amazonia were once thought to reflect frequent village relocation motivated by environmental limitations to growth and permanence [145]. Later, this assumption was questioned: it was estimated that large populations could be sustained permanently in the same location based on agriculture, and social and political, rather than ecological, reasons for fission and relocation were highlighted [39,64,142,146]. As in the case

of village fission, here we reproduce the process of village relocation without attempting to determine its ultimate causes. We initialized the models with permanence times of 10–30 years.

*Leap distance*: Simulations of demic diffusion often evaluate different modes of expansion, distinguishing between continuous models and discrete models where migrants can "jump" over long distances in search for optimal environments–a mechanism known as "leapfrogging" [7,128]. Such movements are frequently evoked to explain maritime expansions through islands or along the coast [7,9], but may happen over land in heterogeneous environments. Long-distance migrations have been reported for lowland South American societies in historical times, sometimes involving hundreds of kilometers in short periods of time [147]. Agriculturists of the *cerrado* allegedly relocated their villages up to 250 km away [148]. The distances involved in such long migrations are close to what has been reported for some Amazonian gatherers' logistic trips [134,149], but appear unusual for settled farmers, and may have been partly made possible by recent depopulation [148]. We initialized the models with two options: not performing "leapfrogging" at all (distance of 0) or leaping over distances of 150–250 km.

In our model, the process of demic diffusion emerges from the interaction between population growth, fission and village relocation, whose rhythms are determined by the parameters described above. Although the detailed dynamics of polyculture agroforestry are not explicitly modeled (which would be impractical at the spatial and temporal scales of the simulations), they are implicit in the dynamics of growth, fission and movement of the villages, as inferred from the ethnographic literature.

For each run, one village is initialized in a hypothetical center of origin. We executed separate models with different centers of origin. We used the earliest dated site(s) of each archaeological culture and the site with the highest *r* in the space-time regressions. The start date was determined from the intercept of the regression, except when the earliest sites did not result in significant correlations, in which case we adopted the median of the calibrated date (Table 2).

The initial village is created with a population at the fission threshold, so that it immediately gives birth to a daughter village, thus beginning the expansion. Every step of the model is equivalent to one year. Over its course, the agents execute the following procedures (Fig 2):

*Population growth*: The new population is computed as:

$$N_t = N_0 + aN_0 \tag{2}$$

Growing exponentially until maximum population density is reached. At that point, if there are unclaimed cells within its catchment radius, the village can establish a new ceiling for maximum population density. Otherwise, if landscape is saturated, the village population is decreased back to maximum density and is not allowed to grow. In practice, the interaction between this procedure and the dynamics of fission produce a curve approximating logistic growth, with slow growth at the beginning due to frequent fission, followed by exponential growth and, finally, stabilizing at the equilibrium level.

*Fission*: When a village population reaches the fission threshold, two criteria are evaluated: (i) whether there are still uninhabited cells in the immediate vicinity of its catchment and (ii) whether the landscape is not already occupied by too many neighbors. The maximum number of neighbors is set at 6 to ensure that, at the beginning of the expansion, villages will attempt to dispose themselves in the landscape more or less regularly following an hexagonal distribution [150]. When those conditions are not fulfilled, if the village is allowed to 'leapfrog', another search is performed for suitable land at the leap distance. To compensate for the effort of migrating, leapfrogging only happens if the attractiveness of the distant environment is higher than the current cell's. If the landscape is completely settled, with no available space to move in

**Table 2. Sites used as geographical origins of each archaeological culture in the simulations with respective start date.**

| Culture | Site | Criteria | Start date (sim BP) |
|---|---|---|---|
| **Saladoid-Barrancoid** | La Gruta | Earliest site | 4591 |
| **Saladoid-Barrancoid** | Corporito | Best correlation | 5133 |
| **Saladoid-Barrancoid**\* | Saladero | Earliest site | 2982 \*\* |
| **Saladoid-Barrancoid**\* | El Mayal 2 | Best correlation | 3741 |
| **Incised-Punctate** | Corozal | Earliest site | 1624 |
| **Incised-Punctate** | Saladero | Best correlation | 1783 |
| **Tupiguarani** | Abraham | Earliest site | 2411\*\* |
| **Tupiguarani** | Bela Vista | Earliest site | 2410\*\* |
| **Tupiguarani** | Grajaú | Best correlation | 2850 |
| **Una** | Gruta do Gentio II | Earliest site | 3513 |
| **Una** | Gruta do Salitre | Best correlation | 4202 |
| **Una**\*\*\* | Loca da Panela | Earliest site | 3073 |
| **Una**\*\*\* | GO-CP-02 | Best correlation | 3128 |

\* Models of Saladoid-Barrancoid expansion using the short chronology for the Orinoco.

\*\* Median of the calibrated date, used because the correlation between dates and distances was not significant.

\*\*\*Models of Una expansion executed without considering the earliest date for Gruta do Gentio II.

the surroundings, and leapfrogging is not possible, the village does not fission and its population is allowed to grow above fission threshold.

*Movement*: If the village has stayed in the same cell longer than maximum permanence time, it attempts to relocate by searching for uninhabited cells in the adjacency of its catchment. As in the fission procedure, if there is no space available in the immediate neighborhood but leapfrogging is an option, another search is performed at the leap distance.

Since our primary aim is not to model social processes that take place under circumstances of landscape saturation or overpopulation (warfare, emergence of hierarchy), but rather to simulate the expansion front, villages whose surrounding landscape is filled and which can no longer grow become inactive.

The model runs until reaching 500 sim BP, the approximate date of European arrival.

**Model evaluation.** We evaluated the results by comparing simulated arrival times with the earliest $^{14}$C dates per spatial bin (calculated from each hypothetical origin using the distances that produced the highest *r in* the regression analysis). For each site, we computed a score equivalent to the probability of the respective simulated year in the normalized calibrated distribution. The scores of all sites were averaged to obtain a measure of model fitness varying from 0 (no match with the calibrated dates) to 1 (all simulated arrival times coinciding with the peak of the empirical calibrated probability distributions).

We employed a genetic algorithm to find the parameter set that would yield the best fit for each simulated cultural expansion. In complex models such as the simulations that we implemented, genetic algorithms are more efficient for optimization than manually exploring the parameter space [151,152]. As in the case of the simulations, the genetic algorithm was implemented in Python (S2 Appendix). We start with a population of genomes (n = 100), i.e. parameter sets for running the models, each initialized with random values for every parameter based on the ranges defined in Table 1. After all models are evaluated, the genomes with the highest fit (n = 40) are selected to "reproduce", originating the population of genomes that will be passed to the next generation. During reproduction, there is a probability (p = 0.8) of crossover, whereby each "parent" contributes only a section of its genome to the offspring. Furthermore, parameters have a chance (p = 0.2) of experiencing mutations every generation. Among

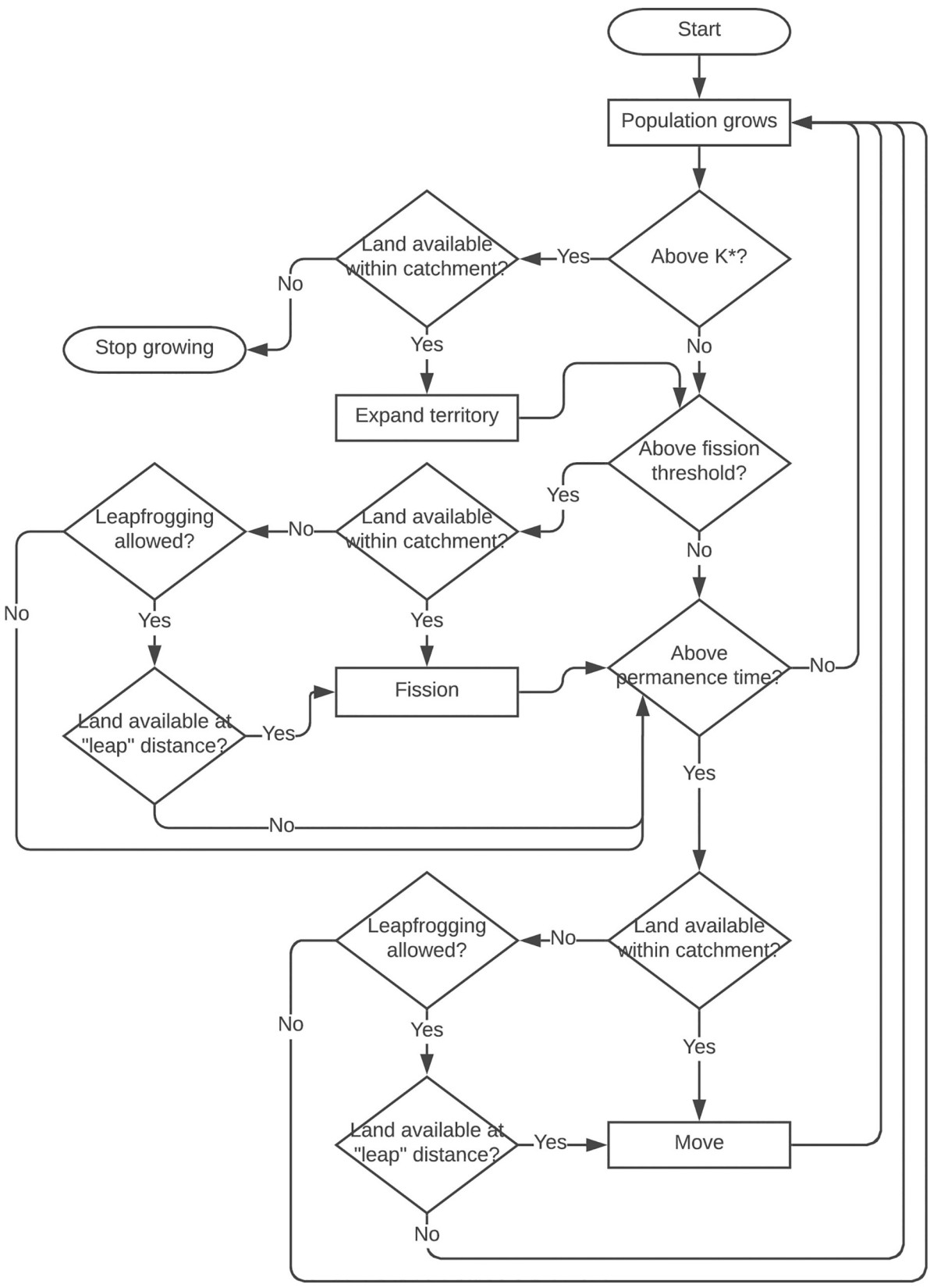

**Fig 2. Model flowchart.**

the genomes with the highest fit, the five best genomes are also preserved and transmitted to the following generation. After a number of generations, the population converges on the optimal parameter set. For a trade-off between computational time and accuracy, we limited execution to 20 iterations, considering that convergence happened fairly early in most runs (S2 Fig).

With a complex set of parameters as the ones selected for our simulations, there was the possibility that different combinations could produce the same or similar results. To account for the possibility of equifinality, we decided to manually control at least for one of the parameters. We have chosen catchment radius, since it was the one with the least variation and with a clear theoretical relationship with another parameter–maximum population density. It is expected that the same population can be maintained by increasing the catchment area while decreasing $K^*$ and vice-versa. Thus, besides running separate models for each geographical point of departure, we executed separate simulations with constant catchment radii of 10, 20 and 30 km.

## Results and discussion

### Regression analysis

Using the earliest site of each archaeological culture as the origin in combination with wide spatial bins (300–500 km) resulted in significant and strong correlations (*r* between -0.83 and -0.87) in nearly all cases (Fig 3 and S3 Table). One exception was the Tupiguarani tradition, for which two sites with similar dates were tested: Bela Vista and Abraham (PA-AT-298), dated 2430 ± 20 $^{14}$C BP and 2410 ± 40 $^{14}$C BP respectively [153,154]. Neither of them produced significant correlations. Another exception was the Saladoid-Barrancoid case when using the short chronology for the Orinoco and Saladero (the earliest site) as origin (Fig 3). Overall, using the short chronology for Saladoid-Barrancoid resulted in poorer fits because many early sites with similar ages would be found at large distances from each other (Fig 3).

Filtering dates through narrower spatial bins resulted in poorer or non-significant correlations (S3 Table). Interestingly, in all cases, the strongest correlations were found when testing for sites other than the earliest. Nevertheless, the sites that yielded the highest coefficients as hypothetical origins tended to be clustered in the same regions and close to the sites with the earliest dates, suggesting that the geographical origin of each archaeological expansion can be identified with reasonable certainty–namely, the Orinoco for Saladoid-Barrancoid and Incised-Punctate, eastern Amazon for Tupiguarani, and central Brazil for Una (Fig 4).

Besides the Tupiguarani, the Una case was found to be problematic. The strong correlations encountered when including the earliest date from the Gruta do Gentio II site [70] were possibly driven by the fact that this date is an outlier, much earlier than any other context belonging to the same tradition (Fig 3). Therefore, we ran a separate set of regressions in a filtered dataset, where that date was removed, which produced significant results (Fig 3 and Table 3).

Although the high values for the *r* coefficient show that a linear fit satisfactorily explains the relationship between dates and distances from origin, it must be noticed that some relationships may be nonlinear. This is most likely caused by changing rhythms of expansion or incomplete radiocarbon data. For example, in the Saladoid-Barrancoid case, the latest date marks a break with the general linear trend, being more recent than expected (Fig 3). A potential correlate is a regular expansion followed by a delay in reaching the final point (southern Amazon), or a lack of dates for the earliest occupation of that region. This delay is not observed using the short chronology for Saladoid-Barrancoid, where the problem seems to be the existence of many sites with similar early ages very far from each other, making it difficult to be modeled as a gradual expansion from a specific origin (Fig 3).

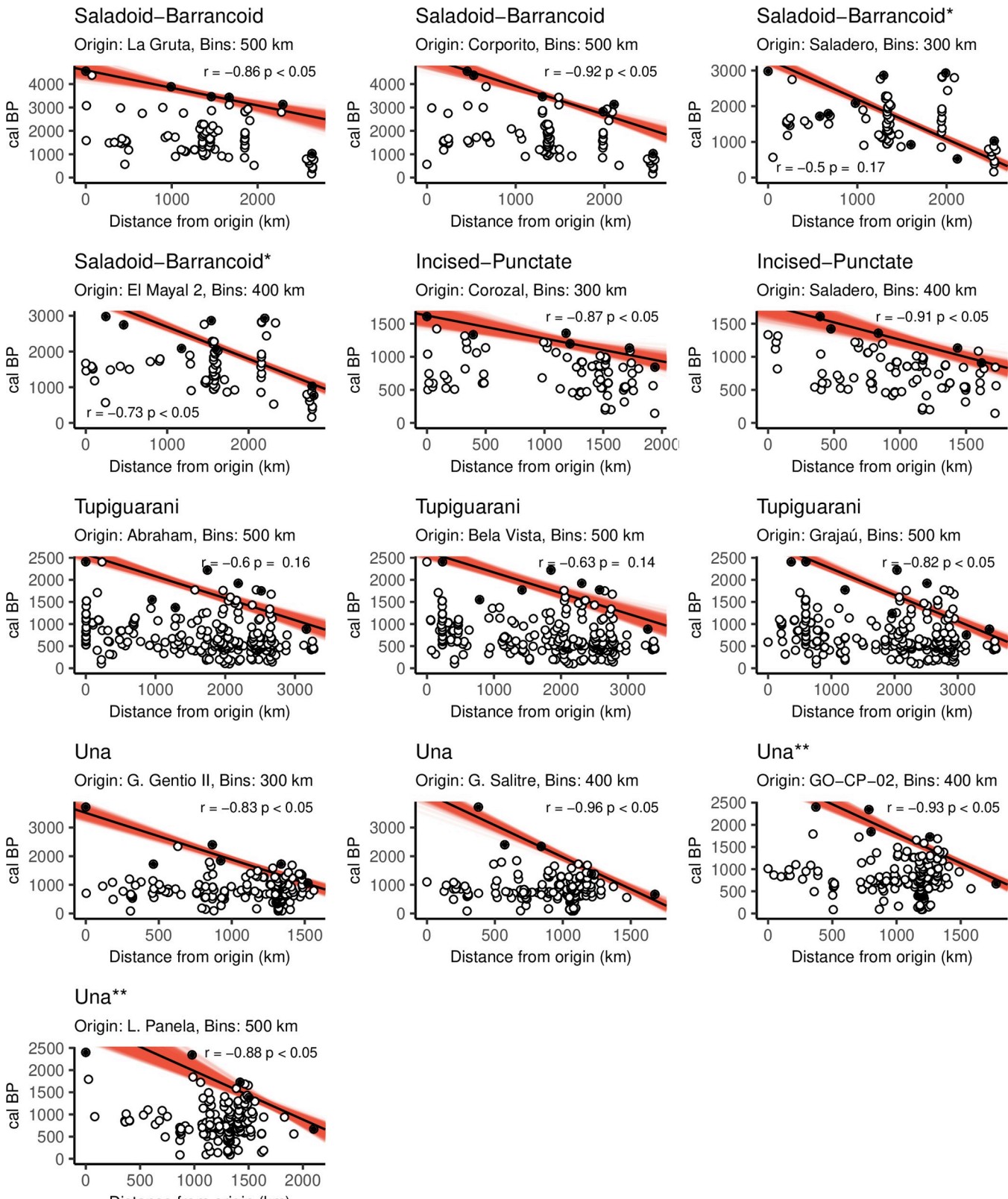

**Fig 3. Scatterplots of dates versus distances from hypothetical origins for each archaeological culture.** Black circles are the earliest dates per spatial bin, whereas white circles are all other dates. Translucent red lines represent RMA fits for 999 iterations drawing single-year values from the calibrated

distributions. The solid black line and the *r* and *p* values refer to the mean of all regressions. *Regressions performed using only dates of the short chronology for the Orinoco. **Regressions performed after excluding the earliest date for Gruta do Gentio II.

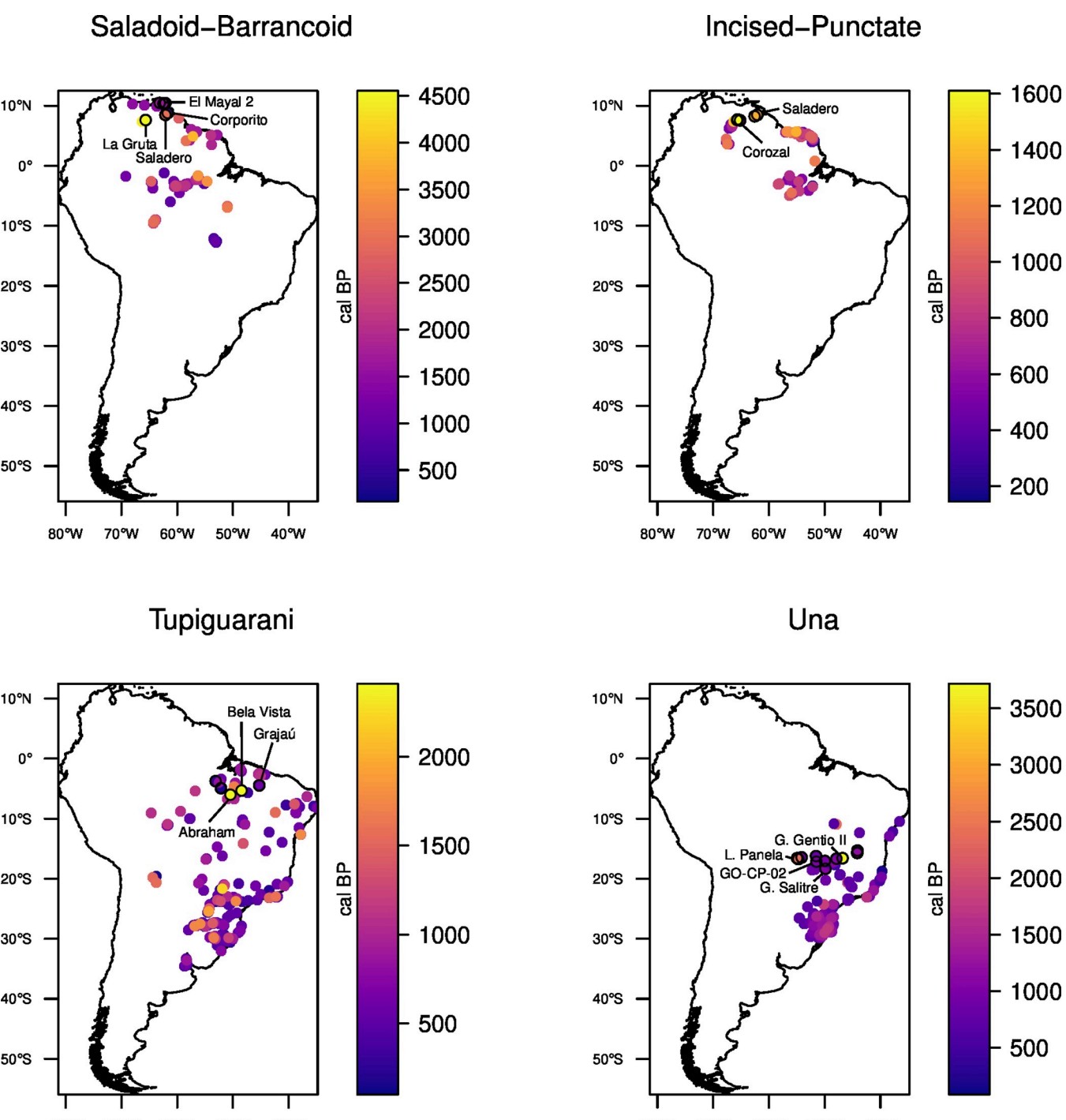

**Fig 4. Distribution of the radiocarbon ages for each archaeological culture.** The earliest sites and the five sites with highest *r* are marked with black outlines. Sites used as potential geographical origins in the simulations are labeled.

**Table 3. Estimated start dates and speeds of advance of each archaeological expansion for different geographical origins, derived from regression intercept and slope, respectively.** Only significant correlations are shown.

| Culture | Origin | Start cal BP | km yr$^{-1}$ | r |
|---|---|---|---|---|
| **Saladoid-Barrancoid** | La Gruta | 4591 ± 168 | 1.33 ± 0.58 | -0.86 |
| **Saladoid-Barrancoid** | Corporito | 5133 ± 199 | 0.82 ± 0.2 | -0.92 |
| **Saladoid-Barrancoid**[*] | El Mayal 2 | 3584 ± 76 | 1.12 ± 0.1 | -0.73 |
| **Incised-Punctate** | Corozal | 1624 ± 83 | 2.88 ± 1.56 | -0.87 |
| **Incised-Punctate** | Saladero | 1783 ± 111 | 1.9 ± 0.84 | -0.91 |
| **Tupiguarani** | Grajaú | 2850 ± 60 | 1.68 ± 0.17 | -0.82 |
| **Una** | Gruta do Gentio II | 3513 ± 116 | 0.61 ± 0.09 | -0.83 |
| **Una** | Gruta do Salitre | 4202 ± 149 | 0.45 ± 0.05 | -0.96 |
| **Una**[**] | GO-CP-02 | 3128 ± 116 | 0.75 ± 0.1 | -0.93 |
| **Una**[**] | Loca da Panela | 3073 ± 276 | 0.91 ± 0.38 | -0.88 |

[*] Models of Saladoid-Barrancoid expansion using the short chronology for the Orinoco.

[**]Models of Una expansion executed without considering the earliest date for Gruta do Gentio II.

The start date of the archaeological expansions and their average speed can be estimated from the regression's intercept and slope, respectively [6,36]. Agricultural expansions elsewhere have been shown to have advanced at a speed of ca. 0.6–1.3 km yr$^{-1}$ [6–8,36,101,128]. Many of the values we encountered were outside that range, suggesting expansions that were much faster or slower, ranging from 0.45 ± 0.05 to 2.88 ± 1.56 km yr$^{-1}$. Although varying the center of origin often led to considerable differences, Una was generally found to be the slowest expansion and Incised-Punctate the fastest. Saladoid-Barrancoid and Tupiguarani were in between, exhibiting values closer to those reported for agricultural expansions in Eurasia (Table 3).

## Simulations

Table 4 summarizes the parameter combinations that yielded the best matches with the archaeological record for each archaeological expansion (a summary of the results for all origins and catchment radii tested can be found in S3 Table). Surprisingly, the scenarios with the highest fits resulted from low population densities, small village sizes and displacement over long distances. This suggests that, although later periods witnessed the emergence of large, permanent settlements with sizable populations in different parts of the Amazon, the first waves of agricultural expansion were led by peoples resembling some of the smallest, most mobile populations of today. As is discussed below, exceptions are not supported by archaeological settlement data.

**Saladoid-Barrancoid.** The best model, using the long chronology and departing from La Gruta (the earliest site), correctly reproduced the overall rhythm of Saladoid-Barrancoid

**Table 4. Parameter sets that produced the best match with the archaeological $^{14}$C dates for each archaeological expansion.**

| Culture | Origin | Maximum population density | Catchment radius | Fission threshold | Leap distance | Permanence | Fitness score |
|---|---|---|---|---|---|---|---|
| **Saladoid-Barrancoid** | La Gruta | 0.26 | 10 | 76 | 170 | 16 | 0.76 |
| **Incised-Punctate** | Saladero | 0.69 | 30 | 86 | 200 | 21 | 0.7 |
| **Tupiguarani** | Grajaú | 0.55 | 30 | 248 | 0 | 10 | 0.42 |
| **Una**[*] | GO-CP-02 | 0.77 | 10 | 142 | 0 | 13 | 0.43 |

[*]Model of Una expansion executed without considering the earliest date for Gruta do Gentio II.

dispersal, except for its southernmost extent (Fig 5), with predicted arrival times in the Guianas and different points of eastern Amazon within the 2σ range of the corresponding calibrated dates (Fig 6). Models departing from Corporito performed considerably worse (S3

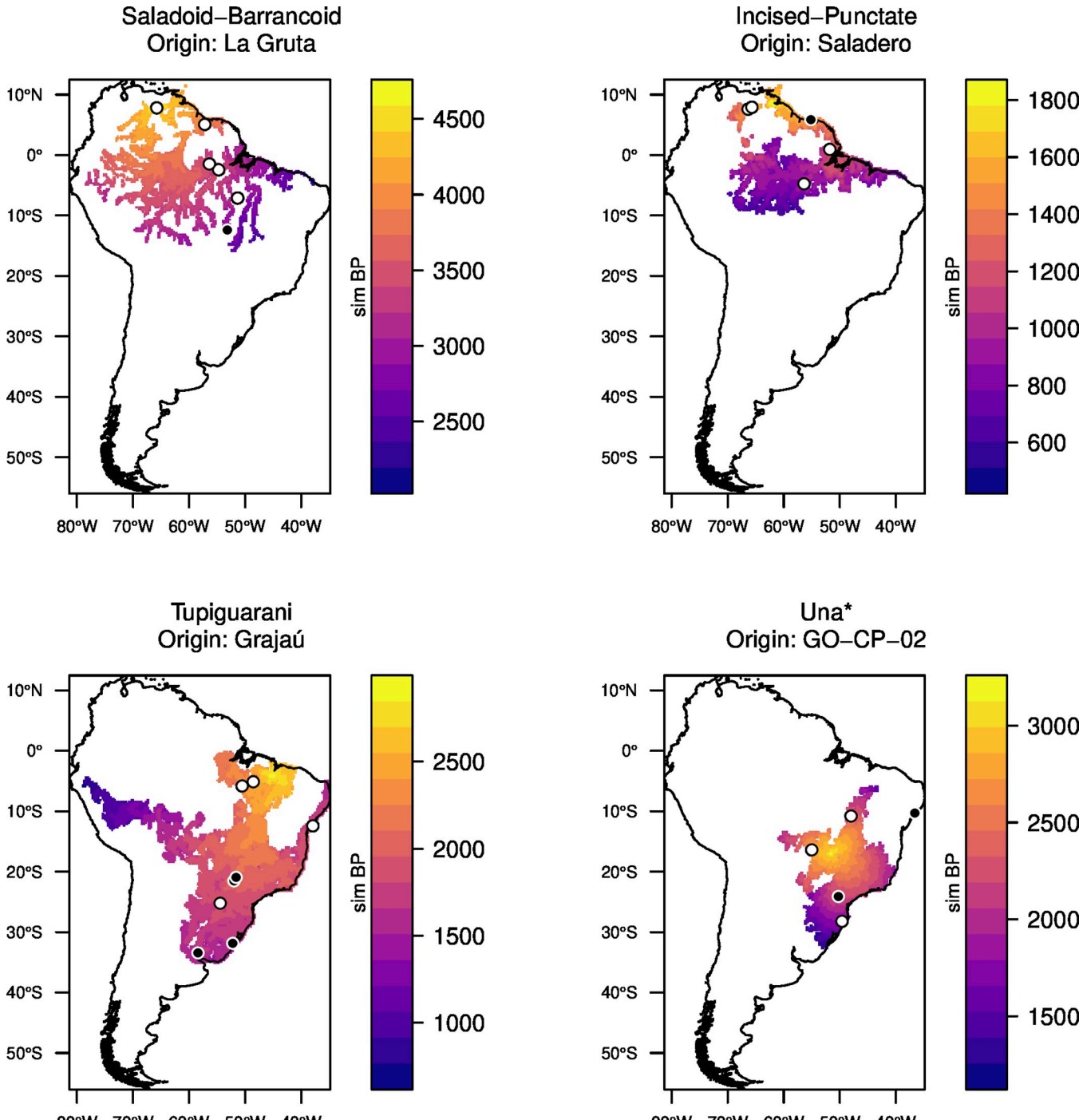

**Fig 5. Results of the simulations executed with the best parameters selected by the genetic algorithm for each archaeological culture.** For the parameters that generated each model, see Table 4. White circles are points where the simulated arrival date (sim BP) is within the 2σ interval of the respective calibrated [14]C age, whereas black circles are points where the simulated arrival date is outside that range. *Model of Una expansion executed without considering the earliest date for Gruta do Gentio II.

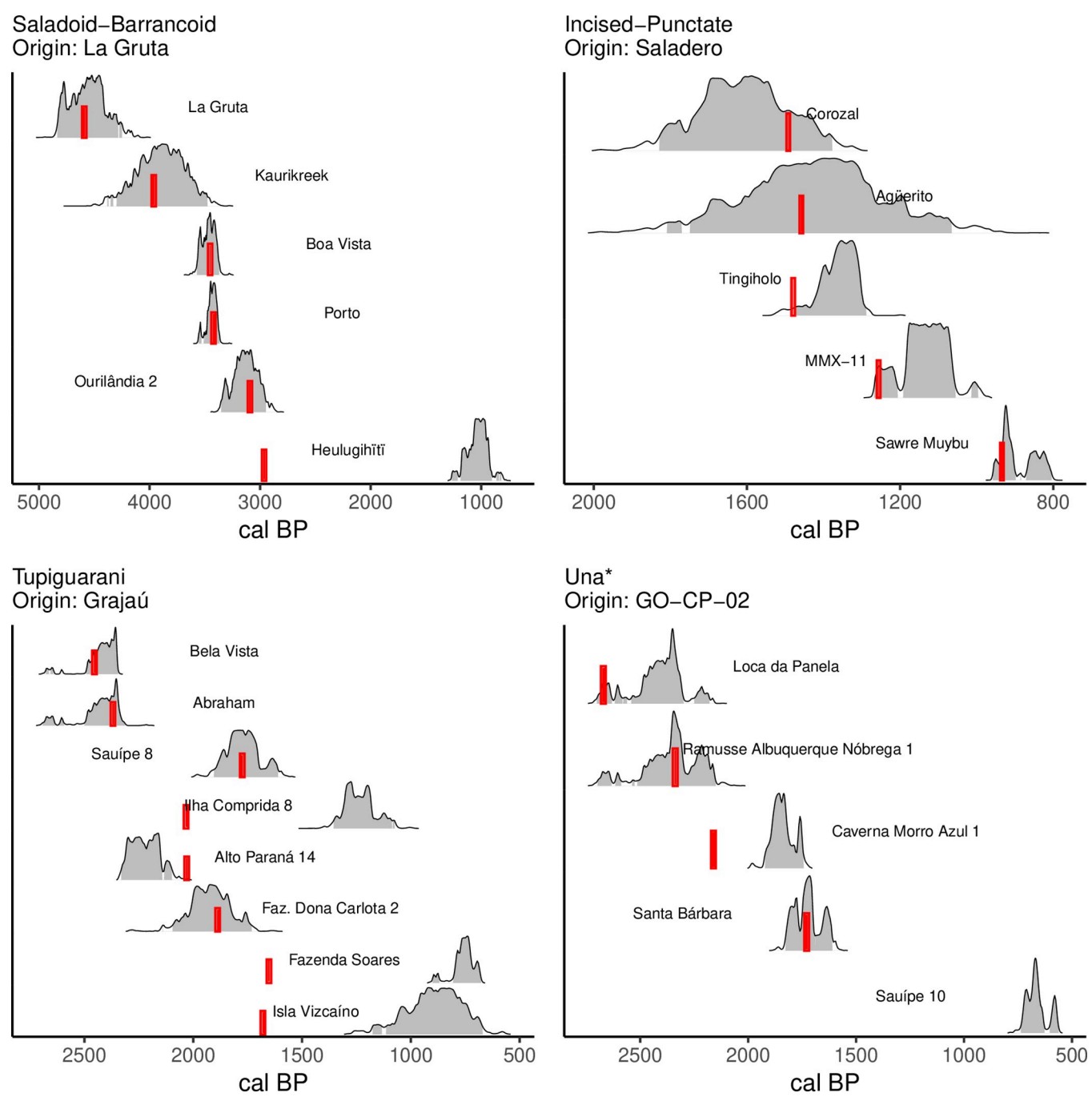

**Fig 6. Comparison between simulated arrival times and dates of earliest archaeological sites for each culture in the models with the best fit.** Shaded areas are the 2σ ranges of the calibrated probability densities of $^{14}$C archaeological dates, accompanied by the respective site name. Red bars represent the simulated arrival time. *Model of Una expansion executed without considering the earliest date for Gruta do Gentio II.

Table, S3 and S10 Figs). Using the short chronology for the Orinoco resulted in poor fits with the archaeological data (S4 and S11 Figs). The parameter set with best performance included low population densities (0.26 individuals km$^{-2}$) and moderate leapfrogging (170 km) (Table 4), but similar results were attained using higher population densities in the absence of leapfrogging (S3 Table and S3 Fig). Even the best model fails to replicate the settlement of

southern Amazon, where the simulated arrival time is 1500–2000 years earlier than seen in the archaeological record (Fig 6, S10 Fig). This should not be seen as a weakness of the model, since it is possible that the dates available for a region do not reflect the earliest occupation [128]. The lag in Saladoid-Barracoid expansion before reaching southern Amazon was already evident in the regression analysis, where the latest date is an outlier in the general linear trend (Fig 3). Earlier dates for that region, closer to the predictions of our simulations but considered intrusive by many archaeologists, should perhaps be reconsidered [39]. Another possibility is that the settlement of southern Amazon did not occur as part of the original Saladoid-Barrancoid expansion, but followed a later wave of advance. This is supported by the distinctiveness of the dated sites for this region. Unlike early Saladoid-Barrancoid sites, which are small ADE occupations without visible architecture, the southern Amazonian sites are large, permanent settlements fortified by multiple ditches [39]. They appear to represent a later development, besides containing a ceramic phase that is quite distinct from the earlier Saladoid-Barrancoid style [21].

Some of the models evaluated using the short chronology for the Orinoco tended to correctly predict arrival times in the northernmost and southernmost extremes of the Saladoid-Barrancoid expansion (S4 and S11 Figs). They failed, however, to capture the timing of the settlement in western, central and eastern Amazon, where arrival times are too late. As noticed in the results of the regression analyses, this is due to the existence of many early dates in those areas, suggesting a nearly simultaneous appearance of the Saladoid-Barrancoid phenomenon if the short chronology is to be accepted.

**Incised-Punctate.** As in the previous case, the simulated Incised-Punctate expansion is in general agreement with the archaeological chronology. The model with the best fit departed from Saladero, which was the site with the best correlation among those considered in the regression analysis (Fig 5), but models departing from Corozal also produced acceptable results (S3 Table, S5 and S12 Figs). The parameter set that produced the best fit included wide catchment radii (30 km) and long leap distances (200 km) (Table 4). Leapfrogging over long distances was indeed present in all models with high fitness (S3 Table). Increasing catchment area and allowing for long-distance movements effectively accelerates the rhythm of expansion, which was expected from the steep slope of the regression analysis of Incised-Punctate dates (Fig 3). In spite of disagreements with archaeological $^{14}$C dates in the region of the Guianas, where simulated arrival times are slightly earlier than expected (Fig 6), the arrival in the lower Amazon was correctly predicted. We consider that the surprisingly rapid Incised-Punctate expansion (possibly as fast as 2.88 ± 1.56 km yr$^{-1}$) can in fact be reproduced using realistic demographic and mobility parameters observed among modern tropical forest farmers.

**Tupiguarani.** Contrary to the previous examples, the spatial patterns of Tupiguarani $^{14}$C dates were not satisfactorily reproduced in any of our simulations. The best results were achieved with the Grajaú site as origin, correctly predicting arrival times in Amazonian sites and the Atlantic coast (Figs 5 and 6). However, even in this scenario, the sequence of dates that reflects the southward movement down the Paraná river [13] is almost completely missed, with simulated dates resulting too early or too late (Fig 6). Other models resulted in poorer fits, correctly predicting some regions at the expense of others, never completely capturing the overall rhythm of expansion (S3 Table, S5 and S11 Figs). For example, in some models departing from the earliest sites, Abraham (PA-AT-298) and Bela Vista, simulated arrival times disagree with the empirical record in nearly all regions except for the extreme south of the Paraná basin (S6 and S13 Figs). Besides the disagreement between the simulated arrival times and the archaeological dates, the parameters that produced the best results are questionable, with a high fission threshold (248 persons) and absence of leapfrogging (Table 4), which contradicts the ethnohistorical data about Tupi-Guarani peoples and their migrations [106,111].

Furthermore, an examination of other parameter sets selected by the genetic algorithm reveal little coherence, varying from scenarios of low mobility with no leapfrogging and long permanence times to scenarios of high mobility with minimum permanence and leap distances at the highest values allowed (S3 Table). Finally, the site that produced the best fit, Grajaú, is located outside the geographical region commonly accepted as the origin of Tupi-Guarani dispersal [95]. Because the disagreements found with the [14]C dates are mainly due to earlier simulated arrival times, it is possible that the archaeological record itself should be reevaluated. In fact, the regression analyses of dates versus distances for a range of hypothetical sites tended to result in weak and non-significant correlations (Fig 3 and S2 Table), suggesting that the available dates are not sufficient to identify a coherent trend. In that sense, some of the early, controversial dates that were excluded from the present analysis [59,93,98] should perhaps be reconsidered.

**Una.** As in the Tupiguarani case, the Una expansion is notably difficult to reproduce. The lowest scores of all simulations were obtained when modeling the Una expansion, in some cases with a complete failure to agree with any of the archaeological dates (S3 Table, S8, S9, S15 and S16 Figs). The best model departed from GO-CP-02 in a scenario that excluded the earliest date from Gruta do Gentio II (Fig 5). When the latter is considered, the models generally fail to predict arrival times in nearly all regions except for the southern Brazilian highlands (S8 and S15 Figs). In addition to the challenge in simulating the overall rhythm of Una expansion, the parameter set that achieved the best results is contradicted by the empirical record. Population densities were the highest of all the best models (0.77 individuals km$^{-2}$) and the fission threshold (142 persons) was higher than for Saladoid-Barrancoid and Incised-Punctate (Table 4). Those parameters are in blatant disagreement with the archaeological settlement data: the earliest Una sites are occupations in rock shelters with ephemeral duration and a low density of material culture [24,70,155]. The challenge we encountered in modeling the Una expansion as a demic diffusion process is in line with previous observations that the earliest sites in the *cerrado* resulted from local transformations among hunter-gatherers [24,55]. It must be noticed that the estimated speed of Una advance derived from the regression analysis (which could be as low as 0.45 ± 0.05 km yr$^{-1}$) is within the range predicted for cultural diffusion processes and observed in parts of Europe where the Neolithic expansion is thought to have proceeded by a mix of demic and cultural processes [10,128].

## Conclusions

The reconstruction of migrations based on the correlation between archaeological cultures and historical societies was a primary concern of South American archaeology in the past, albeit with sparse data that often led to questionable conclusions [32,156]. Over the last decades, however, South American archaeology has accumulated an unprecedented amount of data. Coupled with methodological advances in statistical and computational analyses, this has led to a resurgence of interest in "big data" approaches [27,45,157].

In this paper, we reevaluated late Holocene archaeological expansions in tropical South America, comparing them to cases of demic diffusion elsewhere through the analysis of spatial gradients in radiocarbon dates and by means of computer simulations. We found that some archaeological phenomena could be successfully modeled as demic waves of advance, as is the case of the Saladoid-Barrancoid (using the long chronology) and Incised-Punctate expansions, which gives further credence to their association with the spread of language families, respectively (branches of) Arawak and Carib languages, an association which has long been hypothesized [12,32,77]. Two cautionary notes, however, must be made. First, simulations using the short chronology for the Orinoco failed to model the Saladoid-Barrancoid expansion satisfactorily, mainly due to the existence of several early dates across the Amazon. If correct, these

data would imply that processes other than demic diffusion were behind the dispersal of Sala-doid-Barrancoid material culture–as suggested, for example, by Hornborg's trade model [31]. Second, the Incised-Punctate models depend on the correct identification of the ceramic complexes that should be included in this broadly-defined cultural unit–a definition that may change, for example, if the Koriabo tradition is indeed shown to be an unrelated phenomenon [88]. In this work, we have followed the published literature while recognizing that future data may cast doubt on the expansion of the Incised-Punctate culture and of the Carib languages as a purely demic process.

Other dispersals in our examples were more difficult to simulate as purely demographic processes. In the Tupiguarani case, the source of disagreement appears to be the lack of an identifiable spatial pattern in the distribution of $^{14}$C dates, calling for more research as well as a reappraisal of dates commonly discarded (or uncritically accepted). As for the Una expansion, a strong case can be made for the role of cultural diffusion [24,55]. Thus, the presence of maize and other cultivated plants in the archaeological record of the Una tradition [70–71] may not necessarily imply that agricultural products were the basis of the people's diet and/or the drivers of its expansion, perhaps excluding this archaeological case from the expectations of the FLDH. In any case, a relationship between the Una tradition and the (Macro-)Jê languages should not be discarded, since cultural diffusion could have taken place among pre-agricultural speakers of that language family–as it has, indeed, been suggested in the past [55].

Importantly, the combination of demographic processes for some cultures (Saladoid-Barrancoid, Incised-Punctate) and potentially cultural processes in others (Una) that we see in tropical South America are comparable to the Neolithic expansion in other parts of the globe [100,128], where both situations have been observed. At the same time, we recognize the possibility that to fully understand the majority of archaeological expansions in lowland South America it might require explanations that go beyond purely demic diffusion, especially considering the difficulties in modeling the Tupiguarani expansion or even the Saladoid-Barrrancoid expansion if the short chronology is to be accepted.

Finally, we stress that although an archaeological expansion can be modeled as a demographic process, this does not imply that cultural diffusion did not happen–only that the demic diffusion hypothesis has not been falsified and it can be reasonably held until a cultural diffusion model can be shown to better fit the archaeological data [35]. Thus, in addition to the fact that new $^{14}$C dates may alter the scenarios evaluated here, future work must be directed to simulate scenarios of cultural diffusion that can be compared to our results and possibly improve them, and to extend the analysis to other archaeological cultures of South America. As it has been highlighted before, lowland South America is home to considerable ethnolinguistic diversity [16], and a crucial task for future models will be to assess the role of demic and cultural processes in the emergence of such diversity, beyond the four expansions considered in this work.

## Supporting information

**S1 Table. Archaeological dates and coordinates used in the analysis.**
(ODS)

**S2 Table. Results of the RMA regressions on dates versus distances from all sites considered as potential geographical origins of each archaeological culture.** *Significant at the .05 level. **Significant at the .01 level. ***Regression performed using only dates of the short chronology for the Orinoco. ****Regression performed without the earliest date from Gruta do Gentio II.
(ODS)

**S3 Table. Parameter sets that yielded the highest fitness scores for each simulated expansion considering different points of departure and catchment radii.** *No parameter combination achieved a score higher than 0. **Models of Una expansion executed without considering the earliest date for Gruta do Gentio II.
(ODS)

**S1 Fig. Ecological niches of each archaeological culture modeled using maximum entropy algorithms.** The scale of probability is given in the standard cloglog output of MaxEnt [119].
(TIFF)

**S2 Fig. Performance of the genetic algorithm for each modeled expansion considering different points of departure and catchment radii.** Blue and red lines are, respectively, the average and the maximum fitness of the population of models at each generation. *Models executed using the short chronology for the Orinoco. **Models executed without considering the earliest date for Gruta do Gentio II.
(TIFF)

**S3 Fig. Results of the simulations executed with the best parameters selected by the genetic algorithm for the Saladoid-Barrancoid expansion considering different points of departure and catchment radii.** White circles are points where the simulated arrival date (sim BP) is within the 2σ interval of the respective calibrated $^{14}$C age, whereas black circles are points where the simulated arrival date is outside that range.
(TIFF)

**S4 Fig. Results of the simulations executed with the best parameters selected by the genetic algorithm for the Saladoid-Barrancoid expansion considering different points of departure and catchment radii.** White circles are points where the simulated arrival date (sim BP) is within the 2σ interval of the respective calibrated $^{14}$C age, whereas black circles are points where the simulated arrival date is outside that range. *Models executed using the short chronology for the Orinoco.
(TIFF)

**S5 Fig. Results of the simulations executed with the best parameters selected by the genetic algorithm for the Incised-Punctate expansion considering different points of departure and catchment radii.** White circles are points where the simulated arrival date (sim BP) is within the 2σ interval of the respective calibrated $^{14}$C age, whereas black circles are points where the simulated arrival date is outside that range.
(TIFF)

**S6 Fig. Results of the simulations executed with the best parameters selected by the genetic algorithm for the Tupiguarani expansion considering different points of departure and catchment radii.** White circles are points where the simulated arrival date (sim BP) is within the 2σ interval of the respective calibrated $^{14}$C age, whereas black circles are points where the simulated arrival date is outside that range.
(TIFF)

**S7 Fig. Results of the simulations executed with the best parameters selected by the genetic algorithm for the Tupiguarani expansion departing from Grajaú with different catchment radii.** White circles are points where the simulated arrival date (sim BP) is within the 2σ interval of the respective calibrated $^{14}$C age, whereas black circles are points where the simulated arrival date is outside that range.
(TIFF)

**S8 Fig. Results of the simulations executed with the best parameters selected by the genetic algorithm for the Una expansion considering different points of departure and catchment radii.** White circles are points where the simulated arrival date (sim BP) is within the 2σ interval of the respective calibrated [14]C age, whereas black circles are points where the simulated arrival date is outside that range.
(TIFF)

**S9 Fig. Results of the simulations executed with the best parameters selected by the genetic algorithm for the Una expansion considering different points of departure and catchment radii (continued).** White circles are points where the simulated arrival date (sim BP) is within the 2σ interval of the respective calibrated [14]C age, whereas black circles are points where the simulated arrival date is outside that range. *Models executed without considering the earliest date for Gruta do Gentio II.
(TIFF)

**S10 Fig. Comparison between simulated arrival times and dates of earliest archaeological sites for the Saladoid-Barrancoid expansion considering different points of departure and catchment radii.** Shaded areas are the 2σ ranges of the calibrated probability densities of [14]C archaeological dates, accompanied by the respective site name. Red bars represent the simulated arrival time.
(TIFF)

**S11 Fig. Comparison between simulated arrival times and dates of earliest archaeological sites for the Saladoid-Barrancoid expansion considering different points of departure and catchment radii.** Shaded areas are the 2σ ranges of the calibrated probability densities of [14]C archaeological dates, accompanied by the respective site name. Red bars represent the simulated arrival time. *Models executed using the short chronology for the Orinoco.
(TIFF)

**S12 Fig. Comparison between simulated arrival times and dates of earliest archaeological sites for the Incised-Punctate expansion considering different points of departure and catchment radii.** Shaded areas are the 2σ ranges of the calibrated probability densities of [14]C archaeological dates, accompanied by the respective site name. Red bars represent the simulated arrival time.
(TIFF)

**S13 Fig. Comparison between simulated arrival times and dates of earliest archaeological sites for the Tupiguarani expansion considering different points of departure and catchment radii.** Shaded areas are the 2σ ranges of the calibrated probability densities of [14]C archaeological dates, accompanied by the respective site name. Red bars represent the simulated arrival time.
(TIFF)

**S14 Fig. Comparison between simulated arrival times and dates of earliest archaeological sites for the Tupiguarani expansion departing from Grajaú with different catchment radii.** Shaded areas are the 2σ ranges of the calibrated probability densities of [14]C archaeological dates, accompanied by the respective site name. Red bars represent the simulated arrival time.
(TIFF)

**S15 Fig. Comparison between simulated arrival times and dates of earliest archaeological sites for the Una expansion considering different points of departure and catchment radii.** Shaded areas are the 2σ ranges of the calibrated probability densities of [14]C archaeological

dates, accompanied by the respective site name. Red bars represent the simulated arrival time.
(TIFF)

**S16 Fig. Comparison between simulated arrival times and dates of earliest archaeological sites for the Una expansion considering different points of departure and catchment radii (continued).** Shaded areas are the 2σ ranges of the calibrated probability densities of [14]C archaeological dates, accompanied by the respective site name. Red bars represent the simulated arrival time. *Models executed without considering the earliest date for Gruta do Gentio II.
(TIFF)

**S1 Appendix. Python code for running the simulations.**
(PDF)

**S2 Appendix. Python code for running the genetic algorithm.**
(PDF)

## Acknowledgments

Simulations were run in the MARVIN cluster of the Scientific IT Core Facility, Universitat Pompeu Fabra, Barcelona, Spain.

## Author Contributions

**Conceptualization:** Jonas Gregorio de Souza, Marco Madella.

**Data curation:** Jonas Gregorio de Souza.

**Formal analysis:** Jonas Gregorio de Souza, Jonas Alcaina Mateos.

**Funding acquisition:** Jonas Gregorio de Souza, Marco Madella.

**Investigation:** Jonas Gregorio de Souza.

**Methodology:** Jonas Gregorio de Souza, Jonas Alcaina Mateos.

**Project administration:** Jonas Gregorio de Souza, Marco Madella.

**Resources:** Jonas Gregorio de Souza.

**Supervision:** Marco Madella.

**Validation:** Jonas Gregorio de Souza, Jonas Alcaina Mateos.

**Visualization:** Jonas Gregorio de Souza.

**Writing – original draft:** Jonas Gregorio de Souza, Jonas Alcaina Mateos, Marco Madella.

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
