## [Decision Letter · Decision Letter 0]

27 Jan 2020

PONE-D-19-34282

Archaeological expansions in tropical South America during the late Holocene: assessing the role of demic diffusion

PLOS ONE

Dear Dr. Gregorio de Souza,

Thank you for submitting your manuscript to PLOS ONE. After careful consideration, we feel that it has merit but does not fully meet PLOS ONE’s publication criteria as it currently stands. Therefore, we invite you to submit a revised version of the manuscript that addresses the points raised during the review process.

Please address all the comments before re-submission

We would appreciate receiving your revised manuscript by Mar 12 2020 11:59PM. To enhance the reproducibility of your results, we recommend that if applicable you deposit your laboratory protocols in protocols.io, where a protocol can be assigned its own identifier (DOI) such that it can be cited independently in the future. For instructions see: http://journals.plos.org/plosone/s/submission-guidelines#loc-laboratory-protocols

We look forward to receiving your revised manuscript.

Kind regards,

Peter F. Biehl, PhD

Academic Editor

PLOS ONE

Additional Editor Comments:

Please address all the comments before re-submission

2. We note that Figures 1, 4, 5 and supporting information figures in your submission contain map images which may be copyrighted.

a. You may seek permission from the original copyright holder of Figures 1, 4, 5 and supporting information figures to publish the content specifically under the CC BY 4.0 license. 

Reviewers' comments:

Reviewer's Responses to Questions

**Comments to the Author**

1. Is the manuscript technically sound, and do the data support the conclusions?

Reviewer #1: Yes

Reviewer #2: Yes

2. Has the statistical analysis been performed appropriately and rigorously? 

Reviewer #1: Yes

Reviewer #2: I Don't Know

3. Have the authors made all data underlying the findings in their manuscript fully available?

Reviewer #1: Yes

Reviewer #2: Yes

4. Is the manuscript presented in an intelligible fashion and written in standard English?

Reviewer #1: Yes

Reviewer #2: Yes

5. Review Comments to the Author

Reviewer #1: This is an excellent paper that fully merits to be published PLoS. Human expansions motivated by farming, one of the most important Holocene processes in the history of humankind, has long been understudied in tropical South America in favour of Eurasian perspectives. As rightly pointed out by the author, language expansion models for the Americas have, until now, largely remained hypothetical with little empirical support. The author has compiled an up-to-date impressive radiocarbon dataset (including meticulous radiocarbon hygiene), which has grown rapidly in the last two decades due to the rapid growth of Brazilian archaeology and Amazonia in general. Methods are well described, demographic parameters from the available ethnographic resources of tropical farmers are carefully selected, modelling assumptions are clearly explained and statistics are sound. Conclusions are novel and unexpected showing that there were indeed some cultural traditions experienced demographic expansions in tropical South America while others not. Conclusions are measured and nuanced opening avenues for future research as new data will certainly became available in areas where archaeology has not been carried out yet. Overall, this paper will make a major contribution to language dispersals in tropical South America with larger implications for global prehistory.

Reviewer #2: This is a comprehensive review of data dispersed in different sources, some of them unpublished reports or local publications in Portuguese and Spanish. I commend the authors for their effort in bringing all this information together. It is also a manuscript that addresses the old debate of correlating language and archaeological patterns in lowland South America. The major difference here is that instead of focusing in single language families, the authors employ a continental-scale approach with different language families and archaeological units (phases, traditions etc.). They do so by building simulations from the compiled data set.

I personally thing this is an interesting approach, although I confess I have not much to say about the logic and the procedures that inform the simulation. I also feel that, as much as I am in sympathy with the authors and their effort here, the paper has some problems that need to be dealt with if gets published. Some of these problems are interpretative while others have to do with the data. While the former relate to personal perspectives to approach the data, and are hence minor, the latter need to be addressed or better qualified by the authors.

The authors perspective is that “the vast territories occupied by the largest language families of South America … suggest(…) that sizable demographic expansions may have taken place in pre-Columbian times. Further support is offered by the distribution of 52 archaeological cultures that emerged and expanded over the last 5000 years”. The authors propose that some of theses expansions may result from demographic growth following the adoption of agriculture. To do so they examined four of such language dispersions and correlate them to archaeological patterns to run their simulations and test such assumption.

The four language families addressed in the manuscript – Arawak, Tupi-Guarani, Carib and Ge – have indeed continental scale distributions, mostly the former two. However, such distribution does not need to be explained from a demic diffusion perspective. The idea of correlating demographic expansions with the adoption of agriculture in lowland South America is even older than the concept of demic diffusion and can be credited to Donald Lathrap as the authors rightly acknowledge. That said, the authors themselves concede that the reasoning that provided the backing for such hypotheses – the premise that slash-and-burn itinerant manioc cultivation was a prevailing pattern in the past – has been falsified by recent research. I am in agreement with them when they propose the concept of policulture agroforestry as better way to explain such ancient productive systems, which would have been characterized by the cultivation of short and long-cycle crops, including trees. Davis Harris (2003: 31-32) proposed that “the nutritional potential and expansion capacity of EASs (early agricultural systems) were strongly influenced by the presence or absence of domestic herd animals, cereals, pulses (herbaceous legumes), tree and root crops… Tree crops are nutritionally valuable, especially as a source of vegetable oils, but because they are long-lived perennials their cultivation has been inimical to agricultural expansion. So too has been the cultivation of carbohydrate-yielding root crops, which is commonly complemented with protein obtained by fishing and hunting”. I feel this short description fits really well the case of language dispersals in tropical lowland South America and explains it better than a demic diffusion model. It would be interesting to see how such premises would build into a simulation model.

In the following lines I offer specific comments:

103-104 - “The spread of Saladoid-Barrancoid material culture can be traced to the Orinoco ~4600 cal BP, expanding over most of the Amazon by ~3400-2900 cal BP, except for the southern limits of the basin, which were settled ~1000 cal BP [38,42]”.

223 - The former proposal places the origins of Saladoid ceramics ~4500 cal BP, whereas the latter delays its origins until ~2500 cal BP. The Barrancoid style, on the other hand, is commonly accepted to have begun ~3000 cal BP. Given the lack of consensus, here we maintain the long chronology following recently obtained dates for the Saladero phase, which show that the development of that ceramic style fits poorly in the short chronology [45]. Furthermore, it is easier to conciliate the long chronology with the early dates for Pocó-Açutuba sites in central and lower Amazon [21].

The above assumptions are based in the acceptance of the long-chronology for the Orinoco. The authors are aware of this, but this issue is far from settled. I personally think that a short chronology explains better the data and the patterns there. I may have not noticed, but it would be interesting to run the simulations with a short chronology for the Orinoco as well.

124 – “Incised-Punctate settlement patterns and site architecture are diverse. In the Guianas coast, settlements included habitation mounds surrounded by complexes of agricultural raised fields 125 [50,51].” Here they follow the published literature, but I believe that the presence of Arauquinoid and I-P in the Guiana coast will be reviewed in the light of future research with Koriabo ceramics.

171 – “In spite of the distinctiveness in its settlement patterns, the Una expansion has also been sustained by an agricultural base, as evidenced by the presence of maize in storage facilities since the earliest occupations in rock shelters, together with other domesticated plant remains like manioc and beans in later periods”. I feel that the mere presence of maize in the record does not warrant associating Una expansion with farming. As the authors concede, (661) “as in the Tupiguarani case, the Una expansion is notably difficult to reproduce. Population densities were the highest of all the best models (0.77 individuals km-2 660 ) and the fission threshold (142 persons) was higher than for Saladoid-Barrancoid and Incised-Punctate (Table 4). Those parameters are in blatant disagreement with the archaeological settlement data: the earliest Una sites are occupations in rock shelters with ephemeral duration and low density of material culture [24,69,154]”. I agree with them and believe that the Una case should be removed from a demic diffusion approach.

The final statement reads as: (680) – “In this paper, we reevaluated late Holocene archaeological expansions in tropical South America, comparing them to cases of demic diffusion elsewhere through the analysis of spatial gradients in radiocarbon dates and by means of computer simulations. We found that some archaeological phenomena could be successfully modeled as demic waves of advance, as is the case of the Saladoid-Barrancoid and Incised-Punctate expansions, which gives further credence to their association with the spread of language families, respectively (branches of) Arawak and Carib, an association which has long been hypothesized [12,32,76]. Other dispersals, however, were more difficult to simulate as purely demographic processes. In the Tupiguarani case, the source of disagreement appears to be the lack of an identifiable spatial pattern in the distribution of 14C dates, calling for more research as well as a reappraisal of dates commonly discarded. As for the Una expansion, a strong case can be made for the role of cultural diffusion [24,54], but the reason why it has proceeded much slower than expected remains to be solved in light of recent models that suggest cultural diffusion should proceed faster than demic advance [99]. In any case, a relationship between the Una tradition and the (Macro-)Jê languages should not be discarded, since cultural diffusion could have taken place among pre-agricultural speakers of that language family – as it has, indeed, been suggested in the past [54]. Importantly, the combination of demographic processes for some cultures (Saladoid-Barrancoid, Incised-Punctate) and potentially cultural processes in others (Una) that we see in tropical South America are comparable to the Neolithic expansion in other parts of the globe [99,127] where both situations have been observed.”

Incise-Punctate is a category that probably will be abandoned in the near future as other complexes also potentially related to Carib languages, such as Koriabo, become better known. Hence, maybe the only remaining case for demic diffusion in tropical lowland South America would be Saladoid/Barrancoid/Incised Rim and Arawak. In the light of this, I suggest the authors to rephrase their main argument and structure the ms. as evidence of other processes leading to language dispersals in this part of the world.

I believe that everyone working with this subject may agree that some form of intensification of plant cultivation or agriculture may lay behind these demographic expansions. However, the remarkable fact about tropical lowland South America is the large amount of language families/genealogical units and isolated languages found there: more than 50. Most of those families have a local geographic expression, while some of them may have a wider distribution at a sub-continental scale, such as Pano. The four families addressed in the paper are the ones with largest distribution, but instead of trying to approach those I feel that a major conceptual contribution to archaeology will emerge when one investigates the deep history of the other dozens of families with localized distributions. In other words, the major anthropological question lies in understanding why so much language diversity emerged in the lack of major geographical barriers. This is beyond the scope of this manuscript, of course, but looking at the potential role of demic diffusion tells just one side of the story and does not offer a novel theoretical or factual contribution, as the authors themselves would agree.

6. PLOS authors have the option to publish the peer review history of their article (what does this mean?). If published, this will include your full peer review and any attached files.

Reviewer #1: No

Reviewer #2: Yes: Eduardo Góes Neves

---

## [Author Response · Author response to Decision Letter 0]

4 Feb 2020

Please, see Response to Reviewers file.

---

## [Decision Letter · Decision Letter 1]

14 Apr 2020

Archaeological expansions in tropical South America during the late Holocene: assessing the role of demic diffusion

PONE-D-19-34282R1

Dear Dr. Gregorio de Souza,

We are pleased to inform you that your manuscript has been judged scientifically suitable for publication and will be formally accepted for publication once it complies with all outstanding technical requirements.

With kind regards,

Peter F. Biehl, PhD

Academic Editor

PLOS ONE

Additional Editor Comments (optional):

Reviewers' comments:

Reviewer's Responses to Questions

**Comments to the Author**

1. If the authors have adequately addressed your comments raised in a previous round of review and you feel that this manuscript is now acceptable for publication, you may indicate that here to bypass the “Comments to the Author” section, enter your conflict of interest statement in the “Confidential to Editor” section, and submit your "Accept" recommendation.

Reviewer #1: All comments have been addressed

Reviewer #2: All comments have been addressed

2. Is the manuscript technically sound, and do the data support the conclusions?

Reviewer #1: Yes

Reviewer #2: Yes

3. Has the statistical analysis been performed appropriately and rigorously? 

Reviewer #1: Yes

Reviewer #2: Yes

4. Have the authors made all data underlying the findings in their manuscript fully available?

Reviewer #1: Yes

Reviewer #2: Yes

5. Is the manuscript presented in an intelligible fashion and written in standard English?

Reviewer #1: Yes

Reviewer #2: Yes

6. Review Comments to the Author

Reviewer #1: The minor revisons I proposed have have been thoroughly addressed by the author. I have no further revision to suggest.

Reviewer #2: The authors addressed well the points raised in the review. I believe the revised version will bring a welcome addition of the field.

7. PLOS authors have the option to publish the peer review history of their article (what does this mean?). If published, this will include your full peer review and any attached files.

Reviewer #1: No

Reviewer #2: Yes: Eduardo Góes Neves

---

## [Editor Report · Acceptance letter]

15 Apr 2020

PONE-D-19-34282R1 

Archaeological expansions in tropical South America during the late Holocene: assessing the role of demic diffusion 

Dear Dr. Gregorio de Souza:

I am pleased to inform you that your manuscript has been deemed suitable for publication in PLOS ONE. Congratulations! Your manuscript is now with our production department. 

With kind regards,

on behalf of

Dr. Peter F. Biehl 

Academic Editor

PLOS ONE